# The Role of Gender Differences and Menopause in Obesity-Related Renal Disease, Renal Inflammation and Lipotoxicity

**DOI:** 10.3390/ijms241612984

**Published:** 2023-08-19

**Authors:** Aaron Afonso-Alí, Esteban Porrini, Silvia Teixido-Trujillo, José Antonio Pérez-Pérez, Sergio Luis-Lima, Nieves Guadalupe Acosta-González, Irene Sosa-Paz, Laura Díaz-Martín, Covadonga Rodríguez-González, Ana Elena Rodríguez-Rodríguez

**Affiliations:** 1ITB (Instituto Tecnologías Biomédicas), University of La Laguna, 38200 Tenerife, Spain; aafonsoal@gmail.com (A.A.-A.); teixido.silvia@gmail.com (S.T.-T.); lauradiazmart@gmail.com (L.D.-M.); covarodr@ull.edu.es (C.R.-G.); anarrguez@gmail.com (A.E.R.-R.); 2Research Unit, Hospital Universitario de Canarias, 38200 Tenerife, Spain; 3Department of Animal Biology, Edaphology and Geology, Faculty of Biology, University of La Laguna, 38204 Tenerife, Spain; janperez@ull.edu.es (J.A.P.-P.); ngacostaglez@gmail.com (N.G.A.-G.); alu0101267951@ull.edu.es (I.S.-P.); 4Department of Laboratory Medicine, Complejo Hospitalario Universitario de Canarias, 38200 San Cristóbal de La Laguna, Spain; luis.lima.sergio@gmail.com

**Keywords:** obesity, renal disease, inflammation, lipotoxicity, menopause

## Abstract

The pathogenesis of obesity-related-renal disease is unknown. Menopause can promote renal disease in obese women, but this interaction is unclear. In a previous study, we observed that obese male and female mice developed albuminuria, hyperfiltration, and glomerulomegaly, and these changes were more severe in those obese ovariectomized females. In this study, we also evaluated renal inflammation and lipotoxicity in that animal model. For six months, 43 males and 36 females C57BL6/J mice were randomized to standard diet (SD) or high fat diet (HFD). A group of female animals on SD or HFD was ovariectomized to simulate menopause. We evaluated cytokines: NF-κβ p65, IL-1β, MCP-1, TNF-α, total lipid content, lipid classes, and fatty acid profile in total lipid and individual lipid classes in renal tissue and urine. We found that obese males and females showed higher NF-kβ p-65, TNF-α and MCP-1 in renal tissue, and obese females ovariectomized had higher IL-1β and TNF-α compared with not-ovariectomized. Also, obese animals showed lower proinflammatory and higher anti-inflammatory fatty acids in kidney total lipids, while obese females ovariectomized had a more exacerbated pattern. In brief, obesity induces inflammation and an unbalanced lipidic profile in renal tissue. This pattern seems to be enhanced in obesity after menopause.

## 1. Introduction

Overweight and obesity affect 30–40% of the world’s population. These numbers are expected to increase further in the next decades [1]. The global prevalence of chronic kidney disease (CKD) is about 9% [2]. Obesity and overweight are established risk factors for CKD. However, the prevalence of CKD has not increased in parallel with the pandemic of overweight/obesity, suggesting that not all obese subjects are at risk for renal disease. Hence, identifying those patients in whom obesity induces renal damage is necessary for disease prevention. Obese or overweight subjects within the context of metabolic syndrome (MS) or insulin resistance seem to be at a higher risk for CKD [3,4,5]. MS is particularly prevalent in women after menopause, which may lead to CKD [6]. Thus, gender differences must be considered in the evaluation of the effect of obesity on CKD [6,7]. However, the pathogenesis of renal disease in obese patients and especially in obese post-menopausal women, is not clear.

Challenging mammalian membranes by dietary lipids may lead to robust lipidomic remodeling to preserve membrane physical properties [8]. However, long-term obesity is associated with profound changes in lipid physiology and metabolism, affecting lipid class profiles and fatty acid retailoring of glycerolipids in cellular structures and inducing ectopic lipid accumulation in different organs. Basic research studies observed that these lipid changes may determine dysfunction and damage, particularly in the liver but also in different renal cells, a process called renal lipotoxicity [9,10]. However, the pathways by which lipotoxicity can induce renal damage are complex and remain not completely understood. Inflammation has been associated with lipid deposits, a phenomenon that has been studied deep in liver disease. Lipid deposits induce inflammation, fibrosis, and eventually chronic liver disease [11,12]. Lipid derivatives such as eicosanoids and docosanoids and interconnected cytokines also constitute an interactive network, which seems to play pivotal roles in the onset, development, and resolution of inflammation [13,14]. Whether lipotoxicity and inflammation may determine renal disease and organ failure is still unknown.

Sexual hormones have relevant functions in the kidney. In particular, estrogens reduce insulin-induced sodium reabsorption in tubular cells, attenuate renin–angiotensin–aldosterone system activity, modulate lipogenesis through peroxisome proliferator-activated receptor γ (PPAR-γ), and have protective effects against inflammation and mitochondrial dysfunction via ERα/SIRT1 pathway, among others [15]. The progressive reduction of estrogens in the perimenopausal period may facilitate renal damage in obesity and metabolic syndrome [16,17]. However, the interaction between lipotoxicity and menopause in the induction of renal damage has been rarely investigated.

In a previous study, we found that males and females (ovariectomized or not) animals fed with HFD showed a significant weight increase gain compared with those on SD, being more exacerbated in males than in females. Our main findings were that obese ovariectomized female mice developed insulin resistance, hyperglycaemia, and renal damage evidenced as glomerulomegaly, glomerular hyperfiltration, and increased urinary albumin excretion, despite a similar increase in weight than obese non-ovariectomized female mice. Obese male mice also developed hyperglycaemia, insulin resistance, and hyperfiltration without major renal histological changes [18].

In the present work, we evaluated renal inflammation and lipidomic profile in male and female obese mice, with a special focus on the effect of menopause and obesity.

## 2. Results

### 2.1. Inflammation in Renal Tissue

#### 2.1.1. Inflammation in Male Mice

Obese animals had an increased expression of all inflammatory markers analysed in the renal cortex: NF-κβ-p65, IL-1β, MCP-1, and TNF-α (Figure 1).

#### 2.1.2. Inflammation in Female Mice

Animals on HFD had higher expression of NF-κβ p65, MCP-1, and TNF-α in renal cortex compared with those on SD (Figure 2). Obese ovariectomized animals had an increased expression of inflammatory markers IL-1β and TNF-α compared to female animals on HFD not ovariectomized and higher levels of IL-1β, TNF-α, and MCP-1 than ovariectomized animals on SD (Figure 2).

### 2.2. Lipidomic Analysis in Renal Tissue

#### 2.2.1. Total Lipid Content and Lipid Classes Profile

Total lipids in the kidneys were higher in males on HFD than those in SD (5.6 ± 1.4% vs. 6.9 ± 2.2%, *p* = 0.028) (Figure 3A, Appendix A). No differences were found between females (Figure 3B, Appendix A). Major lipid classes proportions were comparable in all studied groups (Appendix A).

#### 2.2.2. Fatty Acids Profile in Total Lipids

Male mice on HFD had lower levels of 16:0, 18:2 n-6, and 20:3 n-6 and higher levels of 20:2 n-6 and 22:6 n-3 than those on SD (Figure 3C, Appendix A). Females on HFD had higher levels of 20:2 n-6 and 22:6 n-3 and lower of 16:1 n-7 compared with lean females, irrespective of ovariectomy (Figure 3D, Appendix A).

#### 2.2.3. Fatty Acids Profile in Individual Lipid Classes

Male animals: in phosphatidylcholine (PC), obese mice had lower 16:1 n-7, 18:1 n-7, 18:2 n-6, 20:1 n-9, 20:3 n-6, and 22:5 n-6, and higher 22:6 n-3 than lean animals (Figure 4A-left, Appendix A). Phosphatidylethanolamine (PE) of obese mice contained lower 18:1 n-7, 18:2 n-6, 20:1 n-9, 20:3 n-6, and 22:5 n-6, and higher 22:6 n-3 (Figure 4B-left,Appendix A). In phosphatidylinositol (PI), 20:3 n-6 was lower and 20:2 n-6 higher in obese animals (Figure 4C-left, Appendix A). In triglycerides (TAG), except 18:1 n-9, which was higher, major monounsaturated fatty acids (MUFAs) and saturated fatty acids (SFAs) were lower in obese animals compared with leans (Figure 4D-left, Appendix A).

Female animals: in PC, animals on HFD had lower 16:1 n-7, 18:1 n-7, and 20:3 n-6, and higher 22:5 n-3 than those on SD, irrespective of ovariectomy. Obese not ovariectomized had lower 18:2 n-6 and higher levels of 22:6 n-3 than animals on SD. Finally, obese ovariectomized mice had a higher 18:0 than obese not-castrated or castrated animals on SD (Figure 4A-right, Appendix A). In PE, obese animals had lower 20:3 n-6 and 22:4 n-6 and higher levels of 18:0 DMA than those on SD, irrespective of ovariectomy. Obese mice not ovariectomized contained lower 20:0, 20:1 n-9, and 22:5 n-6, and higher 20:5 n-3 and 22:6 n-3 compared to animals on SD (Figure 4B-right, Appendix A). In PI, HFD females showed lower 20:3 n-6 than SD irrespective of ovariectomy (Figure 4C-right, Appendix A). TAG from obese mice showed lower 16:0, 16:1 n-7, and higher 17:0, 18:1 n-9, and 20:2 n-6 than lean animals, irrespective of ovariectomy. Obese castrated animals had high 18:0 and 20:1 n-9 compared with the other groups but lower 15:0 than mice ovariectomized on SD. Obese not ovariectomized showed higher 18:3 n-3 compared with animals obese, castrated, or on SD. Finally, the SD group had a higher 14:0 than ovariectomized animals on the same diet (Figure 4D-right, Appendix A).

### 2.3. Lipidomic Analysis in Urine

#### 2.3.1. Total Lipid Content and Lipid Classes Profile

Urine total lipid content was comparable between groups (Appendix A). In males, obese mice showed higher levels of PC, PE, and cholesterol than lean mice (Figure 5A, Appendix A). Irrespective of ovariectomy, obese females had higher levels of sphingomyelin, PC, and PI than lean-castrated females (Figure 5B, Appendix A).

#### 2.3.2. Fatty Acids Profile

Obese males contained lower levels of 16:0, 18:2 n-6, and 20:0, and higher levels of 18:0 DMA, 18:1 n-9 DMA, 20:4 n-6, 22:5 n-3, 22:6 n-3, and 24:1 n-9 (Figure 5C, Appendix A). In females, obese animals had lower levels of 16:0 and 20:1 n-7 compared to lean mice. Obese ovariectomized animals had lower 18:2 n-6, 18:3 n-6, and 20:1 n-7, and higher 20:2 n-6 compared to lean ovariectomized animals. Finally, obese ovariectomized animals had lower levels of 18:2 n-6 than lean animals (Figure 5D, Appendix A).

## 3. Discussion

We observed that obesity was associated with an increase in inflammatory markers and a lipotoxic profile in renal tissue in male and female animals. These modifications were, in some aspects, more prominent in obese females after menopause. 

Our main finding was that obese animals showed high levels of inflammatory markers in renal tissue, such as NF-κβ p65, TNF-α, and MCP-1. Obesity induces an increase of inflammatory molecules, which can result in the infiltration of cytokines in different organs, i.e., the liver, pancreas, and muscle [10,11]. Our study indicates that obesity-driven inflammation also affects the kidneys. These findings are in line with previous investigations that observed an increase of MCP-1 and TNF-α in the renal tissue of mice fed with HFD for 12 weeks [19,20]. Also, in minipigs fed with HFD for 8 months, Feng et al. observed higher levels of NF-κβ-p65 in renal tissue along with glomerulomegaly, insulin resistance, and hyperglycemia [21]. In humans, higher levels of MCP-1, IL-6, and TNF-α were found in biopsies of patients with diabetic nephropathy or morbid obesity [22]. Inflammation may induce early renal damage, including hyperfiltration, and chronic damage like fibrosis [22,23,24,25]. In this line, in a previous analysis of this model [18], we found hyperfiltration and albuminuria in obese animals, two markers of renal disease possibly related to inflammation. However, our study lasted 6 months, and no relevant fibrosis was found. Further studies, with longer follow-ups, may be needed to test the effect of inflammation in chronic damage.

Both male and female obese animals showed a different lipidomic profile, which was, in some aspects, exacerbated in obese ovariectomized females. In males, total kidney lipid content was higher in obese compared to lean animals. However, a greater difference would have been expected considering the six-fold higher amount of fat in HFD. Also, the distribution of fat was homogeneous, with comparable levels of TAG and cholesterol between obese and lean males, which may indicate a conservative nature of renal tissue and a high lipostatic capacity. Interestingly, female total kidney lipid content was comparable between groups. Lipid infiltration is a major characteristic of obesity, being the fatty liver the main example of this condition [26]. However, our results suggest that this may not be the case for the kidney, particularly in females. Obesity induces an increase in metabolic demand, which in the kidney promotes glomerular hyperfiltration, tubulomegaly, and increased reabsorption of solutes (sodium, glucose, albumin, etc.). The latter requires major amounts of energy [27]. Fatty acids are a major energy substrate in the kidney, which has a more efficient energy production from fat than from glucose [28]. Thus, in obesity, the organ may take extra energy from lipids, limiting intra-renal lipid accumulation. In this line, the HFD was especially enriched with saturated and MUFAs, which were not generally increased in the kidneys of obese animals but even decreased on many occasions. These fatty acids are preferentially oxidized in the context of high-energy demand [28]. Be as it may, this hypothesis requires further investigation.

Obese animals showed an unbalanced lipidomic profile with lower proinflammatory and higher anti-inflammatory fatty acid proportions. Renal tissue of obese animals showed lower 16:0, 16:1 n-7, 18:2 n-6, and 20:3 n-6 but higher 20:2 n-6 and 22:6 n-3. Both 16:0 and several MUFAs can interact with proinflammatory receptors like NF-κβ and PPARs, leading to the production of TNF-α and IL-1β [29]. Moreover, omega-6 fatty acid precursors (18:2 n-6 and 20:3 n-6) can be converted into arachidonic acid (20:4 n-6) and then to pro-inflammatory eicosanoids, i.e., prostaglandins, leukotrienes, and thromboxanes [13]. The reduction of the former two fatty acids may be related to the increase in inflammatory markers observed in renal tissue. Interestingly, high levels of arachidonic acid were found in the phospholipids of kidney tissue regardless of sex and dietary treatment. Considering that the kidney is one of the few organs with constitutive COX-2 and that the presence of prostaglandins is quite large compared to other organs, it makes sense that COX-2 takes arachidonic acid as a substrate, which would justify the high levels of this fatty acid in kidney tissue [30,31,32]. On the other hand, several studies reported that 22:6 n-3 has strong anti-inflammatory properties through the production of pro-resolutive docosanoids such as maresins, protectines, and resolvins [33]. Also, recent evidence indicates that DHA modifies membrane fluidity under high-pressure conditions [34]. So, the up-regulation of this fatty acid might indicate a protective reaction against inflammation or shear stress caused by hyperfiltration and glomerulomegaly. In any case, the increase in this anti-inflammatory fatty acid does not seem to be sufficient to prevent inflammation in kidney tissue. Consistently, these more pro/anti-inflammatory profile responses were also observed in major phospholipids, PC, and PE. 

Some of the overall changes described above were different in obese ovariectomized females, particularly those concerning fatty acids profiles of specific lipid classes. This group presented lower 16:0 and 16:1 n-7 in TAG but higher 18:0 in TAG and PC, as well as a higher content of the PE plasmalogen 18:0 DMA. Also, in PC and PE, the levels of 22:6 n-3 tended to be lower in obese ovariectomized animals, possibly reflecting a more limited or impaired anti-inflammatory response. This latter may suggest that the loss of estrogens in the context of obesity may exacerbate lipotoxicity and inflammation. The kidney has a high density of estrogen receptors (Erα, Erβ, and GPER) [15]. These hormones have an anti-inflammatory effect by interacting with receptors of the NF-kβ system [35], modulate lipogenesis through peroxisome proliferator-activated receptor γ [36], and influence renal hemodynamic by the expression of endothelial nitric oxide (NO) synthase, insulin sensitivity, and sodium reabsorption in proximal tubular cells [15,37]. The lack of these protective effects may promote renal damage in the context of obesity. In this line, estrogen supplementation in ovariectomized animals promoted insulin sensitivity [38], reduced inflammation and accelerated tubular cell regeneration in rats with chronic kidney disease [39], and reduced albuminuria, glomerulosclerosis, and hyperfiltration in a model of diabetic nephropathy [40]. Clearly, this aspect of renal disease deserves future attention.

In some aspects, results from the lipidomic analysis of fatty acids in the urine, such as the presence of higher levels of 22:6 n-3, resembled that of the renal tissue, both in male and female obese animals. Also, higher urinary levels of some phospholipids such as PC, cholesterol, or even PI were found in obese males and females. This finding is intriguing but may indicate the presence of renal cells in the urine since these phospholipids are characteristic of cell membranes [8], suggesting that urinary lipidomics may add valuable information regarding renal damage in obesity. 

Finally, few studies have performed a lipidomic analysis in renal tissue in models of obesity, and no study to our knowledge evaluated the fatty acid profile in individual lipid classes. In mice fed with fat-enriched food with 16:0 and 18:0 (pro-inflammatory) fatty acids, Bonanome et al. found elevated 18:1 n-9 (anti-inflammatory) in kidney tissue [41]. In a model of obese swine fed with HFD for 100 days, Rodriguez-Rodriguez et al. found an unbalanced fatty acid profile with less (pro-inflammatory) 18:2 n-6 and 20:4 n-6 and higher (anti-inflammatory) 18:1 n-9 and 18:3 n-3 fatty acids [42]. These results are in line with our findings and indicate an altered lipid homeostasis that promotes inflammation in kidney tissue. In any case, future studies with longer follow-ups are needed to test the dynamic of lipids in renal tissue and their interaction with inflammation in the context of obesity.

## 4. Materials and Methods

### 4.1. The Animal Model

This is a complementary analysis of a previous study that evaluated obesity-related renal disease in male and female mice [18]. We worked with a model of obesity and metabolic syndrome, the C57BL/6J mice. In brief, 43 males and 36 females C57BL/6J mice were randomized to standard diet (SD) or high-fat diet (HFD) for six months. A group of female animals on SD or HFD was ovariectomized to study the interaction between obesity and menopause in renal damage. Animals underwent several tests during the study, i.e., measured glomerular filtration rate by iohexol, intraperitoneal glucose tolerance test, insulin tolerance test, and 24 h urine collection [18]. For the present study, we evaluated renal inflammation and the lipidomic profile in renal tissue and urine in that model.

### 4.2. Diets

Mice were fed a diet that provided around 60% of the calories from fat (D12492-Research Diets) [43]. The HFD had a 6-fold higher content of total lipids than SD: 4.9 ± 0.2 vs. 29.6 ± 2.0 g lipids/100 g fresh food and a fatty acid content 10 times higher than that of SD: 2.8 ± 0.5 vs. 20.3 ± 1.3 g fatty acid/100 g fresh food. Also, HFD had higher levels of total saturated (SFAs), chiefly 16:0 and 18:0, monounsaturated fatty acids (MUFAs), 16:1 n-9, 18:1 n-9 and 18:1 n-7, and lower levels of polyunsaturated fatty acids (PUFAs), i.e., 18:2 n-6 and 18:3 n-3 than SD (Appendix A).

### 4.3. Inflammatory Markers in Renal Tissue

We evaluated by immunohistochemistry the expression of NF-κβ p-65, IL-1β, MCP-1, and TNF-α in renal tissue. Kidney samples were divided into two parts following the sagittal plane, fixed in 4% paraformaldehyde for 24 h, and embedded in paraffin following standard procedures. Three-micron-thick kidney sections were cut, mounted on slides, deparaffined, and rehydrated. Tissue sections were immersed into Target Retrieval Solution for 15 min at 95 °C for antigen retrieval. Slides were incubated with PBS 1X + 0.1% Triton X-100 for the permeabilization of the membrane and blocked with PBS 1X + BSA 3% + 0.1% azide. Primary antibody was added to the tissues as follows: NF-κβ p-65 (ab16502, 1:200; Abcam, Cambridge, United Kingdom), IL-1β (P420B, 1:100; ThermoFisher, Waltham, MA, USA), MCP-1 (PA5115555, 1:200; ThermoFisher, Waltham, MA, USA), TNF-α (bs-2081R, 1:100; ThermoFisher, Waltham, MA, USA) The slides were incubated with the primary antibody overnight in a humidified chamber at 4 °C. Secondary antibody (Abcam, ab6721, 1:500) was added and incubated for 1 h at room temperature. Finally, sections were stained with DAB and counterstained with hematoxylin. The slides were observed and photographed with an Olympus DP72 camera (Olympus, Tokyo, Japan) fitted to an Olympus DX41 microscope (Olympus). An average of ten to fifteen observation areas per slide were selected in the renal cortex, and the images of these areas were taken under the 10-fold microscope. The positive signals in the renal cortex were identified as the integrated density and were quantified using Image J software version 1.52 (100) (National Institute of Health). An average of the positive signal of all images for each group was determined for analysis.

### 4.4. Lipidomic Analysis in Renal Tissues and Urine

We set up 4 different lipidomic analyses: in renal tissue: total lipid (TL) content, lipid classes (LC), fatty acid profiles of TL, and fatty acid profile of 4 individual major lipid classes: phosphatidylcholine (PC), phosphatidylethanolamine (PE), phosphatidylinositol (PI), and triglycerides (TAG); in urine: TL content, LC and fatty acid profiles of TL. For these analyses we combined thin-layer chromatography (TLC) and gas chromatography coupled to mass spectrometry techniques. The amount of total urine collected allowed only 13 males and 38 females (33 females for lipid classes) to be analysed correctly.

### 4.5. Total Lipid Extraction

A representative wedge of fresh kidney tissue (~50 mg) and 0.1–2 mL of urine was obtained to extract TL by homogenization with chloroform/methanol (2:1, *v*/*v*) [44]. The organic solvent was evaporated under a stream of nitrogen, and the lipid content was determined gravimetrically. TL extracts were stored at −20 °C in chloroform/methanol (2:1, *v*/*v*) containing 0.01% butylated hydroxytoluene (BHT) under an inert atmosphere of nitrogen until further analysis.

### 4.6. Lipid Classes Analysis

Aliquots of 30 µg from TL extracts were used, and LC was separated by high-performance thin-layer chromatography (HPTLC, Merck KGaA, Darmstadt, Germany) in a single-dimensional double-development. 2-propanol/chloroform/methylacetate/methanol/0.25% KCl (5:5:5:2:1.8, *v*/*v*) and hexane/diethyl ether/acetic acid (20:5:0.5, *v*/*v*) were the solvents for polar lipid and neutral fractions, respectively. LC was visualized by charring at 160 °C after spraying with 3% (*w*/*v*) aqueous cupric acetate containing 8% (*v*/*v*) phosphoric acid and quantified in area % in an HPTLC Visualizer (CAMAG, Muttenz, Switzerland) as described by Olsen and Henderson [45].

### 4.7. Fatty Acid Methyl Esters from Total Lipids

1 mg aliquots from the TL extract were transmethylated to obtain methylated fatty acids (Fatty Acid Methyl Esters, FAMEs), adding 1 mL of toluene and 2 mL of 1% (*v*/*v*) sulphuric acid in methanol to the lipid extract. The extracts were incubated in the dark for 16 h in a heating block at 50 °C. After this time, the FAMEs were extracted with hexane: diethyl ether (1:1) and 2% (*w*/*v*) aqueous solution of KHCO_3_. Purification of FAMEs was carried out by thin layer chromatography (TLC, 20 × 20 cm × 0.25 mm), using a mixture of hexane, diethyl ether, and acetic acid (90:10:1, *v*/*v*/*v*) as the mobile phase. The FAMEs were re-dissolved in hexane and stored in glass vials under a nitrogen atmosphere until further determination using a gas chromatograph (TRACE-GC Thermo Scientific, Milan, Italy) [46]. The unequivocal identity of FAMEs was determined using a GC–MS with an Agilent 7890A/7010B (Agilent Technologies^®^, Palo Alto, CA, USA) equipped with an Agilent 7693A autosampler and an HP88 100 m × 0.25 mm i.d × 0.2 µm film thickness column (Agilent Technologies^®^). GC–MS analysis was carried out under the following conditions: injection volume 2 µL, split injection (10:1 split ratio) at 230 °C, GC column temperature starting at 140 °C, 5 min isothermal, 4 °C/min to 240 °C, held for 15 min. The carrier gas was helium (99,9997%) with a constant flow rate of 1 mL/min. The temperature of the transfer line was maintained at 250 °C. Ionization was performed where the impact electrons had kinetic energy levels of 70 eV. The temperatures of the source and quadrupoles were 230 °C and 150 °C, respectively. The MS analysis was carried out in Scan mode with a range of mass between 45 and 450 amu. FAMEs were identified by their retention times and confirmed by comparing their mass spectra with the mass spectra of the NIST library v.2.2 (Agilent Technologies^®^). The selected ion-monitoring mode (SIM) was used for quantification. The profiles of each FAMEs were given in percentage (%) relative area.

### 4.8. FAMEs from Phospholipids and Triglycerides

A subgroup of 36 animals, 6 from each treatment (n = 6), were randomly chosen to determine the fatty acid profile esterified into 4 major lipid classes: PC, PI, PE, and TAG, giving a total of 144 profiles analyzed (36 individuals × 4 FA profiles from lipid classes). To that purpose, after the extraction of TL from kidney tissue, aliquots of 2–4 mg were taken and loaded on TLC (20 cm × 20 cm) silica plates. LC was processed by single-dimensional double-development: polar classes were separated with 2-propanol/chloroform/methyl acetate/methanol/0.25% KCl (25:25:25:10:9 by volume) up to a height of 12 cm in the plate. Neutral fractions were then separated with hexane/diethyl ether/acetic acid (90:10:1 by volume) until the end of the plate. The LC was visualized under UV light after a brief exposure to 0.1% dichlorofluorescein in methanol/water with BHT at 0.01%. Each LC band was scraped from the TLC plates and subjected to direct acid-catalyzed transmethylation on silica for 16 h at 50 °C to obtain FAMEs. The identification of FAMEs was performed in the same way as the identification of total FAMES. The election of the selected lipid classes was based on their abundance in the kidney tissue and their structural and signalling roles in life-limiting activities.

### 4.9. Statistical Analysis

Statistical analysis was performed with GraphPad Prism 8 (GraphPad Software Inc, San Diego, CA) and IBM SPSS Statistics 20 (Chicago, IL). Data are expressed as mean, standard deviation, or median. To compare groups, a 2-way ANOVA was performed, one with an inter-factorial design and one for repeated measures (intra-factorial design). All the animals were included in the analysis except for the fatty acid profile from lipid classes (n = 6 per group).

## 5. Conclusions

Renal inflammation and lipotoxicity may be involved in renal disease in obesity and metabolic syndrome. A more prominent inflammation and lipotoxicity in obese animals after menopause indicated gender differences in obesity-related renal disease. However, the interaction between menopause and renal damage in metabolic syndrome clearly needs further investigation.

## Figures and Tables

**Figure 1 ijms-24-12984-f001:**
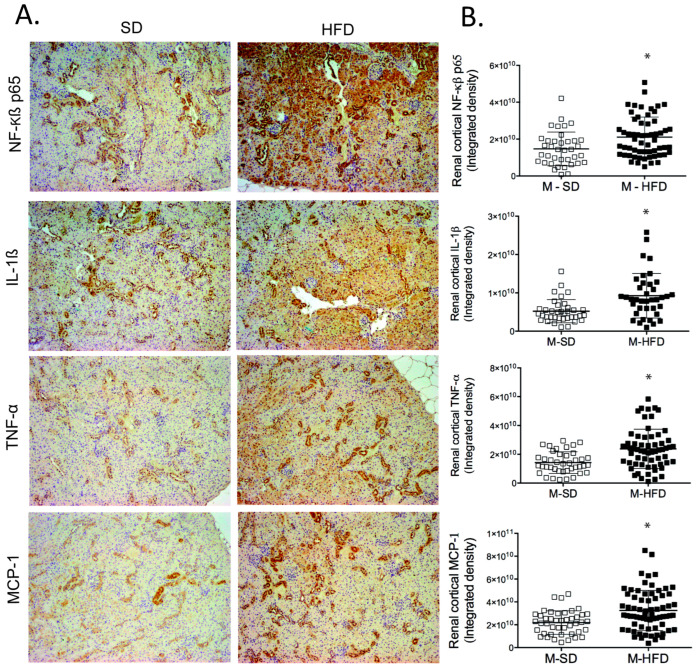
(**A**) Representative images of renal sections of males in SD and HFD; (**B**) Quantitative analysis of each inflammation marker. Data are represented as mean ± standard deviation. *, M-SD vs. M-HFD, *p* ≤ 0.01. M-SD: male standard diet; M-HFD: male high-fat diet.

**Figure 2 ijms-24-12984-f002:**
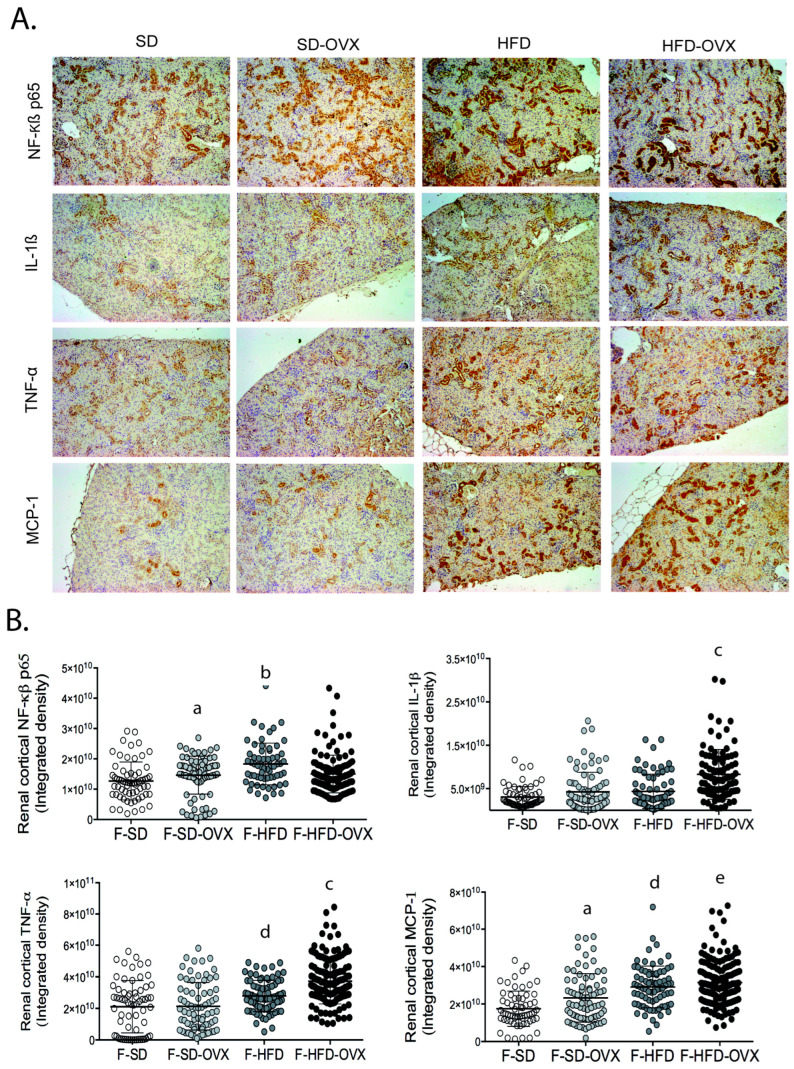
(**A**) Representative images of renal sections of females in SD, SD-OVX, HFD, and HFD-OVX; (**B**) Quantitative analysis of each inflammation marker. Data are represented as mean ± standard deviation. a, F-SD vs. F-SD-OVX, *p* ≤ 0.01; b, F-HFD vs. F-SD, *p* ≤ 0.0001; c, F-HFD-OVX vs. F-HFD and vs. F-SD-OVX, *p* ≤ 0.0001; d, F-SD vs. F-HFD, *p* ≤ 0.01; e, F-HFD-OVX vs. F-SD-OVX, *p* ≤ 0.0001. F-SD: female standard diet; F-HFD: female high-fat diet; F-SD-OVX: female standard diet ovariectomized; F-HFD-OVX: female high-fat diet ovariectomized.

**Figure 3 ijms-24-12984-f003:**
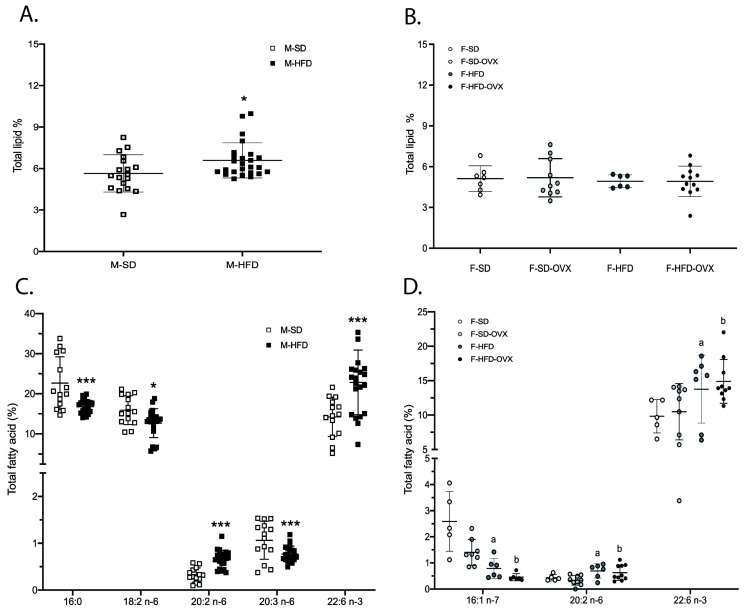
Total lipid and total fatty acids profile in renal tissue of male (**A** and **C**, respectively) and female animals (**B** and **D,** respectively) in relative area%. Mean ± standard deviation. Male differences: * *p* < 0.05, *** *p* < 0.001. Female differences: a = HFD vs. SD and vs. SD-OVX, *p* ≤ 0.05; b = HFD-OVX vs. SD and vs. SD-OVX: *p* ≤ 0.05. M-SD: male standard diet; M-HFD: male high-fat diet; F-SD: female standard diet; F-SD-OVX: female standard diet ovariectomized; F-HFD: females high-fat diet; F-HFD-OVX: female high-fat diet ovariectomized (Appendix A).

**Figure 4 ijms-24-12984-f004:**
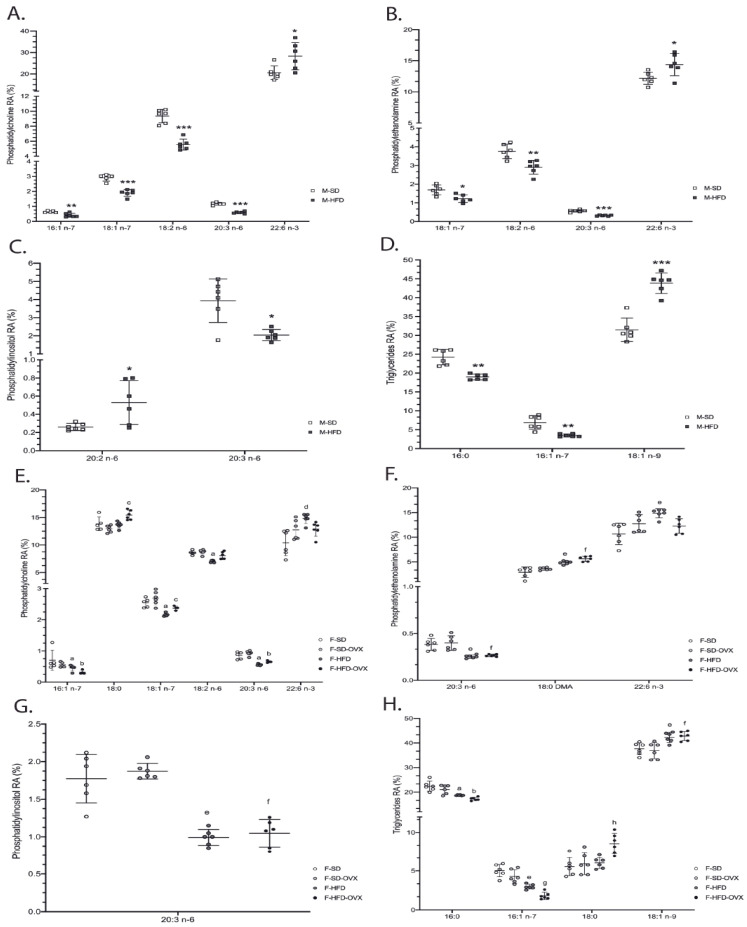
Relative area% (RA%) of fatty acid profile from phosphatidylcholine (**A**,**E**), phosphatidylethanolamine (**B**,**F**), phosphatidylinositol (**C**,**G**) and triglycerides (**D**,**H**) in renal tissue. Data are represented as mean ± standard deviation, n = 6. Males: M-SD vs. M-HFD * *p* ≤ 0.05, ** *p* ≤ 0.01, *** *p* ≤ 0.001. Females: a, F-HFD vs. F-HFD-OVX and vs. F-SD and vs. F-SD-OVX, *p* ≤ 0.05; b, F-HFD-OVX vs. F-HFD and vs. F-SD and vs. F-SD-OVX, *p* ≤ 0.05; c, F-HFD-OVX vs. F-HFD and vs. F-SD-OVX, *p* ≤ 0.05; d, F-HFD vs. F-SD, *p* ≤ 0.05; e, F-HFD vs. F-SD and vs. F-SD-OVX, *p* ≤ 0.05; f, F-HFD-OVX vs. F-SD-OVX and vs. F-SD, *p* ≤ 0.05; g, F-HFD-OVX vs. F-SD-OVX and vs. F-SD, *p* ≤ 0.05 and vs. F-HFD, *p* = 0.052; h, F-HFD-OVX vs. F-SD-OVX and vs. F-SD, *p* ≤ 0.05 and vs. F-HFD, *p* = 0.058. M-SD: male standard diet; M-HFD: male high-fat diet; F-SD: female standard diet; F-SD-OVX: female standard diet ovariectomized; F-HFD: females high-fat diet; F-HFD-OVX: female high-fat diet ovariectomized.

**Figure 5 ijms-24-12984-f005:**
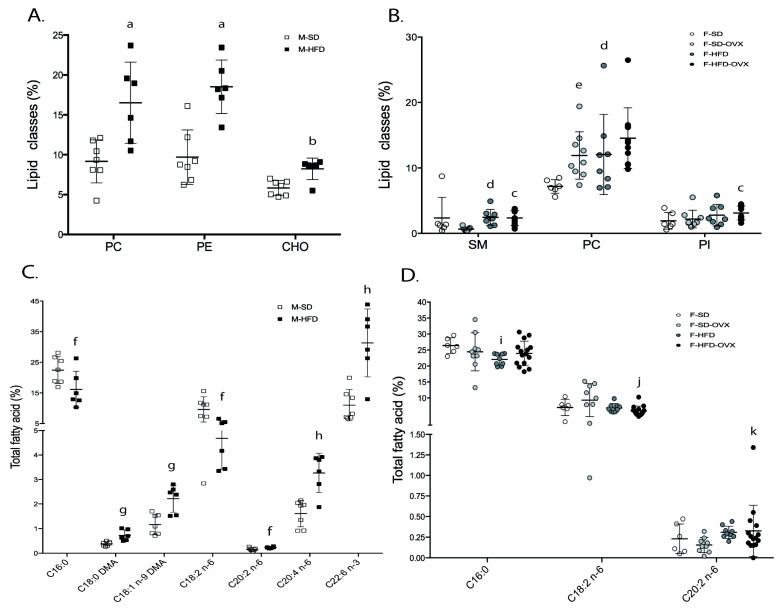
Total lipids and total fatty acid in urine of male (**A** and **C**, respectively) and female (**B** and **D,** respectively) animals. Data are represented as mean ± standard deviation. M-SD: males on standard diet. a, M-HFD vs. M-HFD, *p* ≤ 0.01; b, M-HFD vs. M-HFD, *p* ≤ 0.05; c, F-HFD-OVX vs. F-SD-OVX, *p* ≤ 0.01; d, F-HFD vs. F-SD, *p* ≤ 0.05; e, F-SD-OVX vs. F-SD, *p* ≤ 0.001; f, M-HFD vs. F-SD, *p* ≤ 0.05; g, M-HFD vs. M-SD, *p* ≤ 0.01; h, M-HFD vs. M-SD, *p* ≤ 0.001; i, F-HFD vs. F-SD, *p* ≤ 0.01; j, F-HFD-OVX vs. F-SD-OVX and vs. F-HFD, *p* ≤ 0.05; k, F-HFD-OVX vs. F-SD-OVX, *p* ≤ 0.05. M-SD: male standard diet, M-HFD: male high-fat diet, F-SD: female standard diet, F-SD-OVX: female standard diet ovariectomized, F-HFD: females high-fat diet, F-HFD-OVX: female high-fat diet ovariectomized (Appendix A).

## Data Availability

All the data presented in this study are available in the same manuscript and its Appendix A.

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
