# Peer review of "The Role of Gender Differences and Menopause in Obesity-Related Renal Disease, Renal Inflammation and Lipotoxicity"

_ijms, 2023, doi:10.3390/ijms241612984_

Round 1

Reviewer 1 Report

In this manuscript, the authors investigated the potential role of sex hormones in developing renal inflammation and lipotoxicity. To this end, the inflammation and lipotoxicity in kidney of the male animal, and female animal with or without ovariotomy on different diet were examined and compared. They showed that obese mice showed high levels of inflammatory markers in renal tissue such as NF-κβ, TNF-α and MCP-1. Furthermore obese ovariectomized females had higher IL-1β and TNF-α compared with those without ovariectomy. Additionally, obese mice showed lower proinflammatory fatty acids and higher anti-inflammatory fatty acids in kidney. However ovariectomy caused an exacerbation of these changes. The work provides knowledge advancement to certain extent, but still mainly descriptive. Some major and minor issues are listed below:

Major points: 

1. The finding that renal inflammation is increased upon high fat diet-induced obesity has been reported and lacks novelty.

2. It is meaningless to detect the total expression level of NF-kappa-B p65, since activity of NF-kappa-B pathway is determined by the nuclear translocation of p65, or phosphorylation of p65, but not the total amount of p65.

3. I doubt how the quantification of immunohistochemistry was done, because the staining is a multicolor image (target protein+nuceli staining), so how the signal intensity for the target protein can be quantified in the staining image? Thus realtime pcr and elisa should be used for quantification purpose.

Minor issues:

1. In the last panel of Figure 1B and 2B, “MPC” should be “MCP”.

2. NF-kappa-B is a homo- or heterodimeric complex formed between p65 and p50 subunits. The authors in this manuscript is detecting the expression of p65. Therefore the NF-kappa-B p65, instead of only NF-kappa-B, should be used throughout.

3. There are many typos … such as in line 449, azida should be azide.

The manuscript needs to be checked carefully again to prevent mistakes and typos. 

Reviewer 2 Report

The authors investigated the pathogenesis of obesity related-renal disease regarding renal inflammation and lipotoxicity in male mice and female mice with or without ovariectomy. The results obtained in the present study have a considerable pathological significance in obesity related-renal injuries. However, there is a room for improvement to complete the study.

1.     In the results section, “Preliminary results of kidney function” regarding the author’s previous study need to be moved to the introduction section.

2.     Since there is a possibility that inflammation due to lipidic profile disorders is associated with the oxidative stress, oxidative stress-associated gene expression or immunohistochemistry in renal tissue would be informative.

Round 2

Reviewer 1 Report

All the concerns have been properly addressed.

Reviewer 2 Report

I have no further comment.